# First Clinical Experience of ^68^Ga-FAPI PET/CT in Tertiary Cancer Center: Identifying Pearls and Pitfalls

**DOI:** 10.3390/diagnostics15020218

**Published:** 2025-01-19

**Authors:** Akram Al-Ibraheem, Ahmed Saad Abdlkadir, Ula Al-Rasheed, Dhuha Al-Adhami, Feras Istatieh, Farah Anwar, Marwah Abdulrahman, Rula Amarin, Issa Mohamad, Asem Mansour

**Affiliations:** 1Department of Nuclear Medicine and PET/CT, King Hussein Cancer Center (KHCC), Al-Jubeiha, Amman 11941, Jordan; 2School of Medicine, University of Jordan, Al-Jubeiha, Amman 11942, Jordan; 3Department of Nuclear Medicine, Warith International Cancer Institute, Karbala 56001, Iraq; 4Department of Medicine, King Hussein Cancer Center (KHCC), Al-Jubeiha, Amman 11941, Jordan; 5Department of Radiation Oncology, King Hussein Cancer Center (KHCC), Al-Jubeiha, Amman 11941, Jordan; 6Department of Diagnostic Radiology, King Hussein Cancer Center (KHCC), Al-Jubeiha, Amman 11941, Jordan

**Keywords:** fibroblast activation protein, FAPI, PET/CT, diagnostic pitfalls, therapeutic management, management, TBR, SUVmax, pancreatic adenocarcinoma, hepatobiliary tumors

## Abstract

**Background/Objectives:** Over the past four years, ^68^Ga-fibroblast activation protein inhibitor (FAPI) positron emission tomography/computed tomography (PET/CT) has been established at a tertiary cancer care facility in Jordan. This retrospective study aims to explore tracer uptake metrics across various epithelial neoplasms, identify diagnostic pitfalls associated with ^68^Ga-FAPI PET/CT, and evaluate the influence of ^68^Ga-FAPI PET/CT staging results on changes in therapeutic intent compared to gold standard molecular imaging modalities. **Methods:** A total of 48 patients with biopsy-confirmed solid tumors underwent 77 ^68^Ga-FAPI PET/CT examinations for molecular imaging assessment, encompassing neoplasms originating from the gastrointestinal tract, head and neck, hepatobiliary system, pancreas, breast, and lung. **Results:** Notably, pancreaticobiliary tumors exhibited the highest tracer uptake, with mean maximum standardized uptake values (SUVmax) and tumor-to-background ratios (TBR) surpassing 10. A comparative sub-analysis of ^68^Ga-FAPI PET metrics in 20 treatment-naïve patients revealed a significant correlation between ^68^Ga-FAPI uptake metrics and tumor grade (Spearman’s rho 0.83; *p* = 0.00001). Importantly, the results from ^68^Ga-FAPI PET/CT influenced treatment decisions in 35.5% of the cases, primarily resulting in an escalation of management plans. A total of 220 diagnostic challenges were identified across 88.3% of the scans, predominantly within the musculoskeletal system, attributed to degenerative changes (99 observations). **Conclusions:** This comprehensive analysis highlights the potential significance of ^68^Ga-FAPI PET/CT in oncological imaging and treatment strategy, while also emphasizing the necessity for meticulous interpretation to mitigate diagnostic challenges.

## 1. Introduction

Fibroblast-activation protein (FAP) is predominantly expressed in cancer-associated fibroblasts, which are found in the stroma of 80–90% of all cancers [1]. These fibroblasts produce mediators that influence tumor cells by promoting angiogenesis, migration, and proliferation [2]. The integration of FAP into molecular imaging has progressed rapidly due to its ability to capture the tumor microenvironment and its potential for theranostic applications [3]. Recent reviews have highlighted the association of fibroblast proliferation with various malignancies, making these tumor-associated cells a significant part of the tumor volume [4,5,6]. Large studies have shown that FAP inhibitor (FAPI) positron emission tomography (PET) outperforms ^18^F-fluorodeoxyglucose (FDG) radiotracers, particularly in detecting primary or metastatic liver lesions and cancers of the pancreas, stomach, colon, lung, and ovaries [7,8,9,10]. Peritoneal carcinomatosis has also shown clear superiority over FDG in several studies [11,12,13]. High detection rates for lymph node, bone, lung, and visceral metastases have been observed in many studies, leading to changes in clinical staging. This high detection rate extends to rare neoplasms, including colloid, mucinous, and sarcomatous cancers. Despite variations in uptake intensity across different tumor types, suspicious lesions are easily identified due to a low background signal, resulting in a high tumor-to-background ratio (TBR) with sharp image contrast [9,10]. Efforts are ongoing to explore its potential in various domains, especially with increasing availability and investment in the theranostic market [4].

However, FAPI imaging is not without its challenges and limitations. Diagnostically, the FAPI has suboptimal specificity for detecting malignant tumors, and interindividual variability in FAP expression complicates its widespread use [14,15,16]. Noncancerous conditions involving fibrosis, such as tissue repair, matrix remodeling, and fibrosis in organs, such as the liver, kidney, pancreases, and lung, can also express FAPI [17,18,19,20,21]. Activated fibroblasts are present in conditions such as scar formation, chronic inflammatory diseases, and benign tumors. Therefore, clinicians must consider various conditions when interpreting FAP-targeting tracer images.

In this retrospective observational study, we aimed to explore ^68^Ga-FAPI PET/computed tomography (CT) in various cancer pathologies by examining tumor uptake kinetics and diagnostic pitfalls and measuring the aggregate rate of change in therapeutic intent.

## 2. Materials and Methods

The institutional review board of King Hussein Cancer Center (KHCC) in Amman, Jordan, approved this retrospective study on 24 April 2024 (Registration number: 24 KHCC 72). Informed consent was not necessary because of the study’s retrospective design. The research followed Good Clinical Practice guidelines and the principles of the 1964 Declaration of Helsinki and its subsequent amendments.

### 2.1. Patients

A total of 65 patients who underwent ^68^Ga-FAPI-04 PET/CT scans from October 2022 to March 2024 were retrospectively assessed for inclusion in the study. Patients were only included if they had a confirmed single solid primary malignancy based on histopathology and had undergone a whole-body ^68^Ga-FAPI PET/CT study, resulting in the exclusion of 17 patients. Various clinicopathologic factors, such as gender, age at diagnosis, tumor subtype, primary tumor location, tumor grade, indication for PET/CT imaging, frequency of FAPI PET/CT scans per patient, tumor stage according to AJCC 8th edition [22], and surgical intervention details, were gathered for analysis. Additionally, information from non-FAPI PET/CT studies conducted within a similar timeframe (within 30 days of the ^68^Ga-FAPI PET/CT scan) was also collected when available.

### 2.2. PET/CT Imaging

The PET/CT imaging protocol was implemented using either a Biograph mCT 64 PET/CT system (Siemens, Erlangen, Germany) or a United Imaging uMI 780 PET/CT (Shanghai, China). For the Biograph mCT 64 PET/CT system (Siemens, Erlangen, Germany), reconstructions were performed with and without attenuation correction. CT image acquisition was conducted using a Biograph mCT flow CT scanner (64 slices), and PET images were acquired using Flow Motion technology (Erlangen, Germany). The image reconstruction was performed using the ordered-subset expectation maximization (OSEM) algorithm. Attenuation correction and anatomical localization were achieved through the utilization of low-dose CT without intravenous contrast administration. The slice thickness was 5 mm, and the acquisition process employed a table speed of 1 mm/s, corresponding to a duration of 3 min per bed position. For the uMI 780 PET, images were acquired with 5–6 bed positions (2 min/bed). The PET layer thickness was 3 mm, and all PET images were reconstructed iteratively (OSEM). After reconstruction, post-processing and image fusion were performed using Syngo.via software (version VB40, Siemens, Erlangen, Germany) for image analysis.

### 2.3. PET Radiotracer Injection

For the ^18^F-FDG PET/CT study, patients were required to fast for at least 4–6 h, ensuring that their serum blood sugar levels were below 11.1 mmol/L. PET/CT images were acquired 60 min after the injection of 3–5 MBq/kg of ^18^F-FDG. In studies using ^68^Ga-FAPI, the intravenous injected activity ranged from 1.85 to 2.59 MBq/kg, with imaging performed 40–60 min after radiotracer injection. For the ^68^Ga-DOTATOC scan, 150–200 MBq was injected intravenously, and PET/CT images were acquired 45–70 min post-injection. All procedures adhered to institutional guidelines.

### 2.4. PET/CT Image Analysis

Two experienced nuclear medicine physicians independently performed a visual qualitative assessment and semi-quantitative analysis of all PET/CT scans. Positive tracer-avid lesions were identified by areas of non-physiologic uptake over the background in all images. Positive lesions were further categorized as primary malignancy, non-malignant lesions, distant metastases, or lymph node metastases. Diagnostic pitfalls were characterized as findings that did not pertain to a specific disease or scan objective. Patient histories were meticulously reviewed from medical records, and potential diagnostic challenges were systematically evaluated by two physicians. Patients showing areas of increased uptake levels compared to the background activity were documented in a table using a Microsoft Excel, version 2021 (Washington, DC, USA), spreadsheet, along with relevant etiological information based on their medical background. The region and system of origin corresponding to the increased uptake were also documented for each patient. Semi-quantitative analysis was then performed by comparing the uptake of the radiotracer within the same lesions. The maximum standardized uptake value (SUVmax) was measured for regions of interest (ROIs) defined on the lesion by the two physicians. The tumor-to-background ratio (TBR), defined as the ratio of lesion SUVmax to background SUVmean, was calculated for the malignant lesions and the different radiotracers.

### 2.5. Statistical Analysis

The Shapiro–Wilk test was employed to assess the data distribution. Variables with a normal distribution were presented as the mean ± standard deviation (SD), while those without a normal distribution were expressed as the median and interquartile range (IQR). Categorical variables were represented by percentages (%). To analyze categorical variables, we used either the chi-squared test or Fisher’s exact test. For continuous variables, the Mann–Whitney U test or t-test was applied based on their suitability. The Spearman correlation coefficient, also known as Spearman’s rho, was utilized to investigate the presence of significant relationships among the variables under consideration. A Spearman’s rho value greater than 0.8 signified a strong correlation, whereas values below 0.5 indicated a weak correlation, with values falling between these thresholds implying a moderate correlation. A *p*-value of less than 0.05 (*p* < 0.05) was considered statistically significant. Statistical analysis was performed using Stata software, version 17 (College Station, TX, USA). Appendix A expands upon the core subjects in focus and delineates the statistical methodologies employed to elucidate the significance of the multifaceted analysis.

## 3. Results

This study retrospectively analyzed a total of 48 patients. The majority of the patients (56.2%) were female, with a mean age at diagnosis of 49 ± 15 years. The primary reason for undergoing ^68^Ga-FAPI PET/CT imaging was to evaluate solid malignancies originating in the gastrointestinal tract, followed by head and neck neoplasms, pancreatic neoplasms, hepatobiliary neoplasms, and a small number of patients with tumors in other locations (Table 1).

A total of 77 ^68^Ga-FAPI PET/CT studies were conducted on the 48 patients included in this study. These were performed at a median follow-up time of 13 months (interquartile range of 8–15 months). The main reasons for performing ^68^Ga-FAPI PET/CT were response evaluation, staging, follow-up, and recurrence exclusion. Gold standard molecular imaging modalities were performed in 20 patients, primarily for response evaluation, at a mean follow-up duration of 11.5 ± 5 months using either ^18^F-FDG in 17 patients or ^68^Ga-DOTATOC in 3 patients for various indications, as outlined in Table 2. Appendix A provides baseline demographics, clinical characteristics, and histopathologic profiles of included patients for each explored PET radiotracer.

### 3.1. FAPI Uptake Metrics

In total, 77 ^68^Ga-FAPI PET/CT studies were conducted for the aforementioned indications. Among these, 23 studies (29.9%) did not show any signs of neoplastic disease, while the remaining studies detected neoplastic disease activity through ^68^Ga-FAPI PET/CT. The most significant neoplastic lesions were found in tumors of pancreaticobiliary origin, demonstrating a mean SUVmax of 13 with a mean tumor-to-background ratio (TBR) exceeding 10. This was followed by gastrointestinal neoplasms and other cancer subtypes (Figure 1).

There was no significant correlation between ^68^Ga-FAPI PET metrics (SUVmax or TBR) and other tumor or clinical factors, such as cancer stage, and cancer grade, based on Spearman’s correlation coefficient.

#### 3.1.1. Uptake Metrics in Therapy-Naïve Patients

Given that various therapeutic modalities can affect tumor uptake by altering the tumor microenvironment [23], we sought to perform a sub-analysis of ^68^Ga-FAPI uptake metrics in therapy-naïve patients, totaling 20 in number. The analysis of the maximum SUVmax and TBR distribution in this subgroup confirmed the high uptake properties in patients diagnosed with pancreaticobiliary tumors (Figure 2).

An important observation from this subgroup analysis was that FAPI uptake metrics (both SUVmax and TBR) demonstrated a significantly high correlation with tumor grade (Spearman’s rho 0.83; *p*-value of 0.00001), according to Spearman’s correlation analysis. Other factors, including cancer stage, cancer grade, and BMI, did not show a substantial correlation.

#### 3.1.2. Comparison with Non-FAPI Tracers

A sub-analysis of PET-derived metrics, specifically SUVmax and TBR of neoplastic lesions, between ^68^Ga-FAPI and non-FAPI PET tracers revealed the significant superiority of ^68^Ga-FAPI PET/CT in pancreatic, medullary thyroid neoplasms, and cholangiocarcinoma. However, higher ^68^Ga-FAPI-derived SUVmax and TBR values for gastric adenocarcinoma with a signet ring and colonic adenocarcinoma did not reach statistical significance. Other tumor subtypes examined by both FAPI and non-FAPI modalities, including rare tumors such as appendiceal mucinous neoplasms, pulmonary colloid cystadenocarcinoma, and pulmonary mucinous adenocarcinoma, cannot be analyzed by the Mann–Whitney U test due to singular instance (Table 3).

### 3.2. Impact of ^68^Ga-FAPI PET/CT on Clinical Management

An examination of ^68^Ga-FAPI imaging reports, compared to gold standard molecular imaging techniques such as CT and MRI, revealed that reliance on ^68^Ga-FAPI PET/CT led to changes in therapy intent in 17 patients (35.5%). The majority of these changes involved upscaling the management plan (Figure 3).

Only in a minority of patients (*n* = 3; 6.2%), ^68^Ga-FAPI PET/CT helped exclude and minimize false positives imposed by the comparators (Table 4).

### 3.3. Diagnostic Pitfalls

Out of the 48 patients included, a significant number (*n* = 220) of diagnostic pitfalls were identified in all cases, accounting for 88.3% of all performed ^68^Ga-FAPI PET/CT studies (68 out of 77 studies). The most commonly observed diagnostic pitfalls belonged to the musculoskeletal system and were associated with degenerative processes (Figure 4).

Notably, the most intense ^68^Ga-FAPI-avid non-oncologic findings were related to areas affected by active fibrosis caused by leiomyoma uteri, as detailed in Table 5.

## 4. Discussion

Our retrospective cohort-based analysis identified ^68^Ga-FAPI expression in various cancers, explored its clinical impact, and highlighted diagnostic pitfalls. It provides a comprehensive examination of ^68^Ga-FAPI PET/CT diagnostic pitfalls, diagnostic utility, and its impact on clinical management originating from a tertiary cancer center in Jordan involving Arab patients.

### 4.1. FAPI Uptake Metrics

During the evaluation of ^68^Ga-FAPI uptake metrics, we observed that tumors of pancreaticobiliary origin demonstrated the highest ^68^Ga-FAPI expression, with a mean SUVmax of 13 and a mean TBR exceeding 10. In a previous study, Kratochwil et al. reported that cholangiocarcinoma exhibited average SUVmax values exceeding 12 [9]. The authors noted a high degree of ^68^Ga-FAPI uptake in other cancer subtypes, including esophageal, lung, breast, and sarcoma, all of which exceeded an SUVmax of 10 [9].

Our results demonstrated a higher average SUVmax and TBR for ^68^Ga-FAPI compared to non-FAPI radiotracers across different histopathologic cancer subtypes. However, this difference was statistically significant only for pancreaticobiliary and medullary thyroid neoplasms. Despite not reaching statistical significance, the superiority of ^68^Ga-FAPI PET/CT has been a subject of interest in many of these examined malignancies, including rare cancer subtypes [24]. A recent case report by our team highlighted the detailed capability of FAPI over FDG PET/CT in a single patient with pulmonary colloid adenocarcinoma [25]. This was also addressed in a recent retrospective study by Dendl et al., which examined 55 patients with rare tumor entities, including pituitary and medullary thyroid cancer, demonstrating high levels of SUVmax and TBR, reflected by excellent tumor visualization [24]. Pabst et al. examined 10 patients with cholangiocarcinoma and compared ^18^F-FDG PET/CT with ^68^Ga-FAPI PET/CT [26]. Overall, ^68^Ga-FAPI PET/CT demonstrated superior radiotracer uptake in patients with cholangiocarcinoma. Primary tumor SUVmax was significantly higher in ^68^Ga-FAPI PET/CT compared to ^18^F-FDG PET/CT (14.5 vs. 5.2; *p* = 0.04) [26]. Lin et al. enrolled 61 patients with colonic adenocarcinoma to examine ^68^Ga-FAPI uptake metrics against traditional ^18^F-FDG PET/CT [27]. In primary colorectal cancer lesions, the average TBR of ^68^Ga-FAPI was 13.3 ± 8.9, with an SUVmax of approximately 10 ± 5.4 [27].

A recent meta-analysis highlighted the superiority of ^68^Ga-FAPI compared to ^18^F-FDG regarding uptake metrics. All studies included demonstrated average mean SUVmax values higher than 10 for pancreatic cancer patients. The pooled mean weighted difference displayed significantly higher ^68^Ga-FAPI uptake than ^18^F-FDG (7.51, 95% CI: 5.34–9.67, *p* < 0.001) [28].

#### 4.1.1. Uptake Metrics in Therapy Naïve Patients

In our sub-analysis of 20 therapy-naïve patients, we identified a significant correlation between FAPI expression and tumor grade using Spearman’s correlation analysis. This suggests that ^68^Ga-FAPI PET/CT might correlate with tumor aggressiveness and pathological features. A previous study highlighted the correlation between ^68^Ga-FAPI-04 uptake and aggressive pathological features in 37 therapy-naïve patients with pancreatic ductal adenocarcinoma [29]. FAPI expression was more abundant in poorly differentiated pancreatic neoplasms than in well- to moderately differentiated neoplasms. Tumor SUVmax significantly correlated with tumor size, differentiation, and perineural invasion [29]. Similarly, Shi et al. observed prominent FAPI expression in 75% of primary intrahepatic HCC lesions, with higher expression in poorly differentiated forms, during their examination of 25 patients with intrahepatic lesions [30].

#### 4.1.2. Comparison with Non-FAPI Tracers

^68^Ga-FAPI PET/CT is now more widely offered to patients with pancreatic, hepatobiliary, and gastrointestinal malignancies [31,32,33,34,35]. The gold standard pan-tumor molecular imaging agent ^18^F-FDG appears less useful for these tumor subtypes due to lower expression [31,32,33,34]. Many previous studies have outlined the superior capabilities of ^68^Ga-FAPI PET/CT in these malignancies. In pancreatic adenocarcinoma, for example, several studies have highlighted the superior capability of ^68^Ga-FAPI PET/CT over ^18^F-FDG PET/CT, particularly in changing the therapy plan by providing more accurate staging results [36]. A similar observation was reported in hepatobiliary malignancies, where significantly altered overall staging and higher ^68^Ga-FAPI expression over ^18^F-FDG PET/CT were noted in patients with hepatobiliary cancers, including cholangiocarcinoma [37]. Furthermore, ^68^Ga-FAPI PET/CT was more effective than ^18^F-FDG PET/CT in detecting primary lesions and recurrences in colorectal cancer [38]. Its evaluation of metastatic liver lesions was far superior to the results obtained from ^18^F-FDG PET, as per a recent study conducted by Shangguan et al. [38].

### 4.2. Impact of ^68^Ga-FAPI PET/CT on Clinical Management

Another vital target of this study was the evaluation of the clinical impact of ^68^Ga-FAPI PET/CT. While tumor uptake metrics are important and have been heavily explored in the past five years, the exploration of ^68^Ga-FAPI’s impact on therapy plans and decisions remains understudied. Our study advocated for the significant clinical role of ^68^Ga-FAPI PET/CT in providing futuristic insight and guidance for treatment decisions, shaped by its high tumor uptake, reflected by SUVmax and TBR levels. This endorses the reliability of ^68^Ga-FAPI PET/CT staging and its clinical impact, recently highlighted in two major studies [39,40]. In the cohort under investigation, five patients underwent treatment escalation subsequent to the identification of peritoneal carcinomatosis that had been overlooked by other imaging modalities. This underscores the significant value of ^68^Ga-FAPI PET/CT in this context [41]. Furthermore, ^68^Ga-FAPI PET/CT demonstrated superior performance compared to alternative imaging techniques in the detection of local disease recurrence, affecting five patients. Additionally, four patients experienced treatment escalation following the identification of distant metastatic lesions revealed by ^68^Ga-FAPI PET/CT. These findings highlight the diagnostic utility of ^68^Ga-FAPI PET/CT in the detection of recurrent, metastatic, and peritoneal involvement in various epithelial neoplasms [42].

Our study supported a previous one conducted by Koerber et al. [39], which found a 42% rate of change in therapy intent affecting their study cohort of 226 patients with various solid malignancies, predominantly resulting in upstaging the patient and upgrading the therapy plan. The second and more recent study, by Kosmala, reported similar positive results, affecting the treatment decision in one-fourth of the 32 examined patients with various solid malignancies [40].

### 4.3. Diagnostic Pitfalls

As we become more familiar with this novel diagnostic agent and its theranostic capabilities, it becomes essential to study and highlight its non-oncologic findings to avoid misinterpretation, which can impact patient care. Therefore, we aimed to examine the list of non-oncologic findings in our patients to highlight common observations. Notably, the majority of the studies conducted have a fair share of ancillary findings, as detailed previously. Based on our findings, the most prevalent non-oncologic findings were related to the musculoskeletal system and attributed to ongoing degenerative processes. The quest for reviewing and examining diagnostic pitfalls has become more abundant recently, after several reports of various inflammatory, autoimmune, degenerative, and benign diseases [15,43]. A collective effort to examine these pitfalls was represented in a recent systematic review by Bentestuen et al., which identified a total of 1178 papers, of which 108 were eligible [14]. A total of 2372 ^68^Ga-FAPI-avid nonmalignant findings were reported, with the most frequent being uptake in the arteries, e.g., related to plaques (*n* = 1178, 49%). Endorsing our findings, ^68^Ga-FAPI uptake was also frequently related to degenerative and traumatic bone and joint lesions (*n* = 147, 6%) or arthritis (*n* = 92, 4%) [14]. Moreover, Kessler et al. conducted a pictorial analysis of various pitfalls encountered in 91 patients and found that the majority of these pitfalls belonged to musculoskeletal degeneration, substantiating our findings [44]. Inflammatory uptake can also occur and has been reported in many previous reports and images [45,46]. Even stent-induced uptake due to recent stent introduction can lead to intense diffuse ^68^Ga-FAPI PET/CT [47]. Sometimes, these can mask or mimic the primary disease of interest, introducing diagnostic challenges and complexities [21,48]. Therefore, examining and reviewing such pitfalls is essential to improve reporting and avoid misinterpretation.

### 4.4. Study Limitations

Our research is subject to several limitations. Our analysis was conducted retrospectively at a single cancer center and included a small group of patients with diverse oncologic malignancies at various stages of disease. Despite these limitations, our study represents the first and only investigation of ^68^Ga-FAPI PET/CT in Jordan and the Arab world. Furthermore, this research provides the first head-to-head comparison of PET-derived parameters in medullary thyroid cancer patients, comparing ^68^Ga-FAPI with ^68^Ga-DOTATOC. Appendix A offers further insights into the key differences between our study and a selection of previous investigations from other parts of the world.

## 5. Conclusions

This study suggests potential benefits of ^68^Ga-FAPI PET/CT in assessing a wide range of solid neoplasms, particularly pancreaticobiliary malignancies. As per our results, ^68^Ga-FAPI PET/CT appeared to provide higher PET-derived uptake metrics compared to non-FAPI tracers and influence treatment decisions in a substantial portion of cases. However, the limited sample size, inclusion of heterogeneous histopathologic cancer subtypes, and the retrospective, single-center nature of this study warrant cautious interpretation. The high prevalence of diagnostic pitfalls observed in our study, especially in the musculoskeletal system, underscores the importance of considering patient history for accurate interpretation. Despite its diagnostic challenges, there is a growing interest in utilizing ^68^Ga-FAPI PET/CT for non-cancer-related conditions to monitor diseases associated with fibrosis.

## Figures and Tables

**Figure 1 diagnostics-15-00218-f001:**
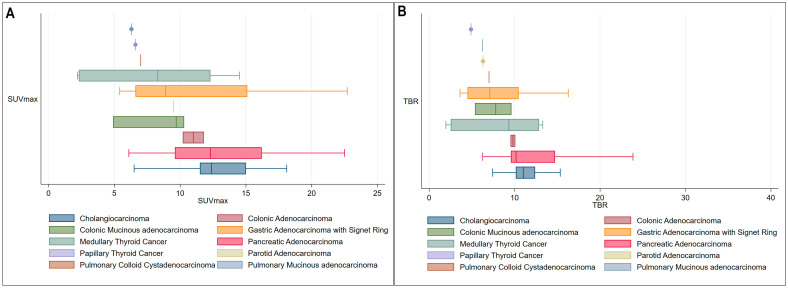
(**A**) Box plots demonstrating the distribution of average maximum standardized uptake values (SUVmax) for various cancer subtypes. (**B**) Box plots demonstrating the distribution of average tumor-to-background ratios (TBRs) for various cancer subtypes.

**Figure 2 diagnostics-15-00218-f002:**
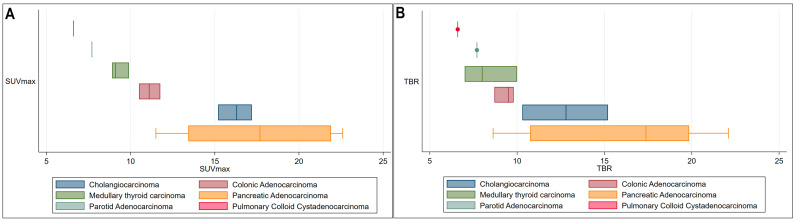
(**A**) Box plots demonstrating the distribution of average SUVmax for various cancer subtypes in therapy naïve patients. (**B**) Box plots demonstrating the distribution of average TBR for various cancer subtypes in therapy naïve patients.

**Figure 3 diagnostics-15-00218-f003:**
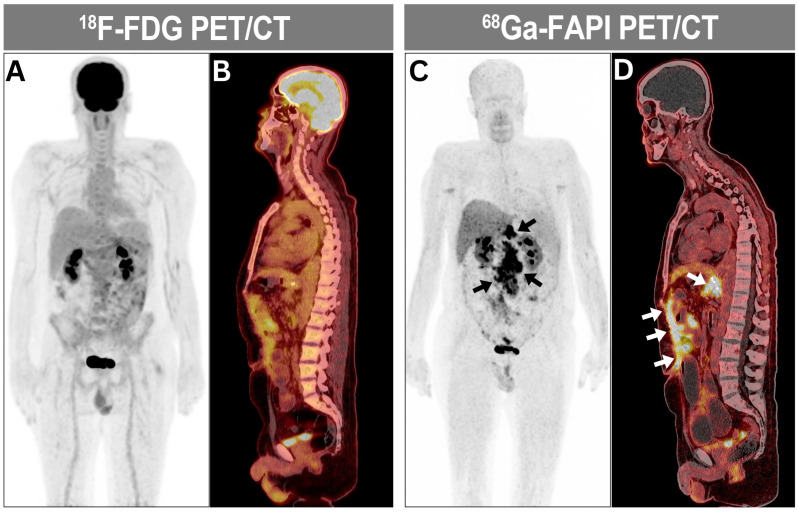
A 74-year-old male patient with histopathology-proven gastric adenocarcinoma with signet ring following subtotal gastrectomy and abdominal lymphadenectomy and adjuvant chemotherapy. (**A**,**B**) At the end of chemotherapy, the maximum intensity projection (MIP) and sagittal views of ^18^F-FDG PET/CT were unremarkable for residual and/or recurrent disease. (**C**,**D**) On the contrary, ^68^Ga-FAPI MIP and sagittal views of ^68^Ga-FAPI PET/CT acquired 18 days later demonstrated interval development of intensely ^68^Ga-FAPI-avid peritoneal deposition consistent with peritoneal carcinomatosis (arrows), prompting an upgrade in the patient’s treatment plan.

**Figure 4 diagnostics-15-00218-f004:**
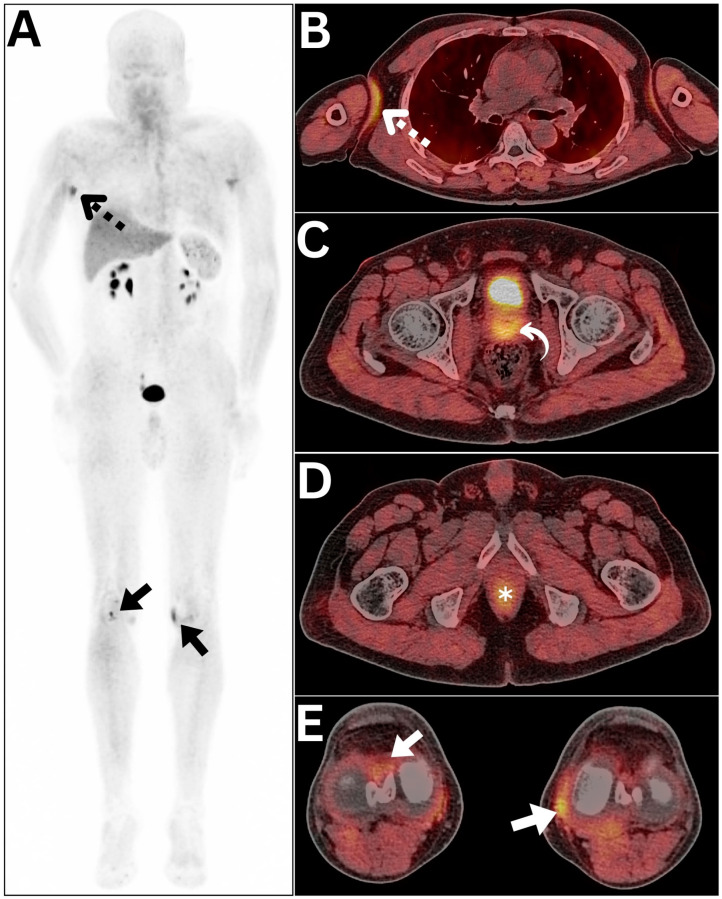
(**A**) A multitude of diagnostic pitfalls were encountered during the ^68^Ga-FAPI PET/CT evaluation of a 61-year-old male patient with colonic mucinous following right hemicolectomy and adjuvant chemotherapy as evident in the MIP image (annotations). (**B**) An axial chest PET/CT image revealed a mildly ^68^Ga-FAPI-avid focus involving the right axillary fold, likely of non-specific nature (dotted arrow). (**C**) An axial pelvic PET/CT image exposed diffuse ^68^Ga-FAPI localization at the site of an enlarged prostate due to benign prostatic hyperplasia (curved arrow). (**D**) An axial pelvic PET/CT image identified sphincteric ^68^Ga-FAPI localization at the anorectal junction (asterisk). (**E**) An axial lower extremity PET/CT image demonstrated ^68^Ga-FAPI localization within both knee joints, which can be best ascribed to the degenerative process (arrows).

**Table 1 diagnostics-15-00218-t001:** Baseline demographics, clinical characteristics, and histopathologic profiles for the included patients.

**Demographics**			
**Age (in Years**)			
Mean ± standard deviation			49 ± 15
**Gender**			***n* ^1^ (% ^2^)**
Female			27 (56.2%)
Male			21 (43.8%)
**Histopathologic Characteristics**			
**Tumor Site**			***n* (%)**
Gastrointestinal			19 (39.6%)
Head and neck			11 (22.9%)
Pancreas			9 (18.7%)
Hepatobiliary			6 (12.5%)
Lung			2 (4.2%)
Breast			1 (2.1%)
**Primary Neoplasm**	***n*, %**	**H-Stage ^3^ (*n*)**	**Grade (*n*)**
Pancreatic adenocarcinoma	9, 18.7%	IV (*n* = 4); III (*n* = 4); II (*n* = 1)	P ^4^ (*n* = 6); M ^5^ (*n* = 3)
Gastric adenocarcinoma with signet ring	6, 12.5%	III (*n* = 4); IV (*n* = 2)	P (*n* = 6)
Medullary thyroid cancer	6, 12.5%	IV (*n* = 4); II (*n* = 2)	P (*n* = 5); W ^6^ (*n* = 1)
Cholangiocarcinoma	6, 12.5%	IV (*n* = 4); II (*n* = 1); III (*n* = 1)	P (*n* = 4); W (*n* = 1); M (*n* = 1)
Colonic adenocarcinoma	5, 10.4%	III (*n* = 3); II (*n* = 1); IV (*n* = 1)	M (*n* = 4); P (*n* = 1)
Appendiceal mucinous Neoplasm	4, 8.3%	IV (*n* = 3); II (*n* = 1)	W (*n* = 4)
Colonic mucinous neoplasms	4, 8.3%	II (*n* = 2); III (*n* = 1); IV (*n* = 1)	W (*n* = 4)
Papillary thyroid cancer	3, 6.2%	III (*n* = 1), II (*n* = 1); I (*n* = 1)	W (*n* = 2); P (*n* = 1)
Parotid adenocarcinoma	2, 4.2%	IV (*n* = 2)	P (*n* = 2)
Pulmonary mucinous neoplasm	1, 2.1%	II (*n* = 1)	W (*n* = 1)
Pulmonary colloid cystadenocarcinoma	1, 2.1%	III (*n* = 1)	W (*n* = 1)
Invasive lobular carcinoma	1, 2.1%	III (*n* = 1)	P (*n* = 1)
**Cancer Staging**			***n* (%)**
Stage I			1 (2.1%)
Stage II			10 (20.8%)
Stage III			16 (33.3%)
Stage IV			21 (43.7%)
**Therapeutic Interventions Prior to first conducted PET/CT**			***n* (%)**
Therapy-naïve			20 (41.6%)
Surgical tumor resection			16 (33.3%)
Chemotherapy			11 (23%)
Radiotherapy			1 (2.1%)

^1^ *n*: Number of patients; ^2^ %: Percentage; ^3^ H-Stage: Histopathologic staging according to American Joint Committee on Cancer 8th edition [22]; ^4^ P: Portly differentiated; ^5^ M: Moderately differentiated; ^6^ W: Well-differentiated.

**Table 2 diagnostics-15-00218-t002:** Table highlighting the main indications of PET/CT studies.

**Indication of ^68^Ga-FAPI PET/CT (77 Studies in Total)**	***n* ^1^ (% ^2^)**
Response evaluation following chemotherapy	28 (36.4%)
Staging	25 (32.4%)
Follow-up	17 (22.1%)
Evaluation following surgery	4 (5.2%)
Excluding recurrence	2 (2.6%)
Evaluation following radiotherapy	1 (1.3%)
**Indication of Non-FAPI PET/CT (21 Studies in Total)**	***n* (%)**
Staging	12 (55%)
Response evaluation following chemotherapy	8 (40%)
Response evaluation following radiotherapy	1 (5%)

^1^ *n*: Number of patients; ^2^ %: Percentage.

**Table 3 diagnostics-15-00218-t003:** A sub-analysis of PET-derived metrics, specifically SUVmax and TBR of neoplastic lesions, between fibroblast activation protein inhibitor (FAPI) and non-FAPI PET tracers.

	**PET/CT ^1^ Parameters (Median and Interquartile Range)**					
**Parameter**	**SUVmax ^2^**			**TBR ^3^**		
	**FAPI ^4^**	**FDG ^5^**	** *p* ** **-Value**	**FAPI**	**FDG**	** *p-* ** **Value**
CCA ^6^	15.2 (10.2–18.1)	3.5 (2.5–6.2)	0.02	13.8 (10.3–15.1)	2.7 (1.7–3.6)	0.01
PDAC ^7^	18.5 (9.5–21.1)	4.8 (3.2–7.7)	0.01	10.1 (9.2–14.3)	4.1 (2.9–6.1)	0.006
GCA with signet ring ^8^	7.9 (6.6–22.2)	4.1 (2.2–9.4)	0.27	6.4 (4.4–16.2)	3.1 (2.9–9.3)	0.31
Colonic Ca ^9^	11 (10.1–11.8)	8.9 (6.5–11.3)	0.39	12.7 (10.1–13.6)	8.5 (7.9–10)	0.12
**Parameter**	**SUVmax**			**TBR**		
	**FAPI**	**DOTATOC ^10^**	** *p* ** **-Value**	**FAPI**	**DOTATOC**	** *p-* ** **Value**
MTC ^11^	8.4 (7.1–9.3)	3.7 (2.2–6.5)	0.04	7.8 (6.6–9.1)	2.9 (2.1–4.6)	0.04

^1^ PET/CT: Positron emission tomography/computed tomography; ^2^ SUVmax: Maximum standardized uptake value; ^3^ TBR: Tumor-to-background ratio; ^4^ FAPI: Fibroblast activation protein inhibitor; ^5^ FDG: Fluorodeoxyglucose; ^6^ CCA: Cholangiocarcinoma; ^7^ PDAC: Pancreatic ductal adenocarcinoma; ^8^ GCA with signet ring: Gastric adenocarcinoma with signet ring; ^9^ Colonic Ca: Colonic adenocarcinoma; ^10^ DOTATOC; 1,4,7,10-tetraazacyclododecane-1,4,7,10-tetraacetic acid Tyr3-octreotide; ^11^ MTC: Medullary thyroid cancer.

**Table 4 diagnostics-15-00218-t004:** Summary of key remarks on the impact of ^68^Ga-FAPI PET/CT on clinical management.

Patients (*n* ^1^)	Details	Comparator	Impact on Management
2	Detection of peritoneal carcinomatosis otherwise not reported	^18^F-FDG ^2^ PET/CT ^3^	Upscaled
1	Detection of peritoneal carcinomatosis and extra-abdominal lymph nodes otherwise not reported	CT	Upscaled
1	Detection of metastatic mesenteric lymph nodes otherwise not reported	CT	Upscaled
2	Detection of peritoneal carcinomatosis otherwise not reported	CT	Upscaled
2	Detection of hepatic metastases	MRI ^4^	Upscaled
2	Detection of recurrence	MRI	Upscaled
2	Detection of primary lung lesion otherwise not depicted through FDG PET/CT molecular imaging	^18^F-FDG PET/CT	Upscaled
1	Detection of primary medullary thyroid cancer and metastatic lesions otherwise not radiotracer-avid on ^68^Ga-DOTATOC imaging	^68^Ga-DOTATOC PET/CT	Upscaled
1	Detection of small metastatic lymph nodes and hepatic lesions in a patient with high thyrocalcitonin levels but negative CT and ultrasound	CT and neck Ultrasound	Upscaled
1	Ruling out neoplastic involvement of the skeleton falsely reported as suspicious by CT	CT	Downscaled
1	Obviated the need for biopsy as the suspicious omental nodules on CT were not showing FAPI activity	CT	Downscaled
1	Accurate restaging, as the local recurrence and metastatic lesions were not concerning on MRI	MRI	Downscaled

^1^ *n*: Number of patients; ^2 18^F-FDG: ^18^F-Fluorodeoxyglucose; ^3^ PET/CT: Positron emission tomography/computed tomography; ^4^ MRI: Magnetic resonance imaging.

**Table 5 diagnostics-15-00218-t005:** Overview of the observed diagnostic pitfalls, including their relevant system, location, and etiology.

**Musculoskeletal System (99 Observations)**		
**Etiology**	**Location**	**SUVmax ^1^ Range**
Degenerative joint uptake	Multi-vertebral	2.5–4.6
	Bilateral acromioclavicular joint	2.5–4.5
	Sternoclavicular joint	2.6–3.3
	Bilateral knee joints	2.3–4.1
	Peripatellar uptake	3–4.2
Reactive uptake	Insulin injection site	2.5–3
	Gluteal intramuscular injection	2.7–4.3
Muscular overactivity	Bilateral paraspinal muscle	2.6–3.5
	Intercostal muscle overuse due to tachypnea	3.9–4.2
Heterotopic ossification	Bi-pelvic myositis ossificans	5.5–7.9
Sphincteric activity	Anorectal junction	2.9–4.1
**Abdomen and Pelvis (57 Observations)**		
**Etiology**	**Location**	**SUVmax Range**
Benign tumors	Leiomyoma uteri	5.1–10.3
Hyperplasia	Benign prostatic hyperplasia	4.8
Interventional	Peri-regional uptake surrounding recently introduced biliary or pancreatic ducts	3.5–5.3
Surgery-induced	Cesarian section scar	3.8–5.5
	Abdominal wall incision	4.1–5.2
Inflammatory	Pancreatitis	3.9–6.1
	Cholangitis	3.1–5.8
	Inflamed Paraumbilical hernia	4.4
**Thorax (25 Observations)**		
**Etiology**	**Location**	**SUVmax Range**
Inflammatory	Pleural effusion	2.5–3.6
	Pneumonia	3–4.7
Surgery-induced	Cardiac pacemaker implant	3.3
Benign tumor	Breast fibroadenoma	5.1
Radiotherapy-induced	Radiation pneumonitis	3.6
**Skin (20 Observations)**		
**Etiology**	**Location**	**SUVmax Range**
Non-specific	Bilateral axillary folds	2.1–3.9
**Head and Neck (19 Observations)**		
**Etiology**	**Location**	**SUVmax Range**
Dental procedures	Peri-gingival uptake	2.2–3.6
Inflammatory	Sinusitis	2.9–3.6
Interventional	Peri-tubal uptake surrounding recently introduced NG ^2^ tube	2.6–3.5

^1^ SUVmax: Maximum standardized uptake value; ^2^ NG: Nasogastric.

## Data Availability

The data presented in this study are available on request from the corresponding author. The data are not publicly available owing to privacy.

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
