# Peer review of "First Clinical Experience of 68Ga-FAPI PET/CT in Tertiary Cancer Center: Identifying Pearls and Pitfalls"

_diagnostics, 2025, doi:10.3390/diagnostics15020218_

Round 1
Reviewer 1 Report
Comments and Suggestions for Authors the radiopharmaceutical under study (FAPI) is undoubtedly very interesting and with excellent prospects. The scientific community is starting to talk about it and its value will be seen over time. The study undoubtedly highlights the characteristics of this molecule, however it is very heterogeneous, brings together many neoplastic pathologies and leads to somewhat confusing conclusions. I probably would have focused attention on a single neoplastic pathology or on a single district (for example head-neck or gastrointestinal neoplasia). Furthermore, the work also involves other tracers (see FDG) but the comparison with FAPI is not very clear. The characteristics of both radiopharmaceuticals must be clearly highlighted. There is also talk of therapeutic application. Perhaps I would have excluded it by discussing it in an ad hoc paper. In conclusion, the work is adequately written but collects very inconsistent data. It is a collection of case studies that should be homogenized by talking about one topic at a time.Author Response
Dear Reviewer 1
Thank you very much for your insightful comments. Kindly note that we have chosen to subdivide each question with number labeling to facilitate the review process and answer each question accordingly.
Below are the answers to your respectful review points.
- The radiopharmaceutical under study (FAPI) is undoubtedly very interesting and with excellent prospects. The scientific community is starting to talk about it and its value will be seen over time. The study undoubtedly highlights the characteristics of this molecule; however, it is very heterogeneous, brings together many neoplastic pathologies and leads to somewhat confusing conclusions. I probably would have focused attention on a single neoplastic pathology or on a single district (for example head-neck or gastrointestinal neoplasia).
- Thank you for your insightful remarks. The conclusion section have been rewritten to acknowledge current limitations and avoid any confusion (gray text, lines 376-385). Our study minimized heterogeneity by using separate distribution box plots and Mann-Whitney U tests for each histopathologic subtype. Overall analysis of uptake metrics and modality comparisons was avoided to prevent confusion. This approach is evident in the box plots, Figure 1, Figure 2, Table 1, and Table 3. Supplementary tables (Table S1 & S2) further clarified our adopted mode of analysis for each of examined topic.
- Thank you for your insightful remarks. The conclusion section have been rewritten to acknowledge current limitations and avoid any confusion (gray text, lines 376-385). Our study minimized heterogeneity by using separate distribution box plots and Mann-Whitney U tests for each histopathologic subtype. Overall analysis of uptake metrics and modality comparisons was avoided to prevent confusion. This approach is evident in the box plots, Figure 1, Figure 2, Table 1, and Table 3. Supplementary tables (Table S1 & S2) further clarified our adopted mode of analysis for each of examined topic.
- Furthermore, the work also involves other tracers (see FDG) but the comparison with FAPI is not very clear. The characteristics of both radiopharmaceuticals must be clearly highlighted.
- Table 3 was updated to include the radiotracer of interest employed in comparison against FAPI (gray highlights, lines 210-217). We’ve also created supplementary table (Table S2) to provides Baseline demographics, clinical characteristics, and histopathologic profiles of included patients for each explored PET radiotracer (Cited in lines 166-168; gray highlights, and supplied as a supplementary excel sheet).
- Table 3 was updated to include the radiotracer of interest employed in comparison against FAPI (gray highlights, lines 210-217). We’ve also created supplementary table (Table S2) to provides Baseline demographics, clinical characteristics, and histopathologic profiles of included patients for each explored PET radiotracer (Cited in lines 166-168; gray highlights, and supplied as a supplementary excel sheet).
- There is also talk of therapeutic application. Perhaps I would have excluded it by discussing it in an ad hoc paper.
- Your valuable input is much appreciated. We find the idea of conducting an ad-hoc analysis on the clinical application of FAPI PET/CT quite compelling. However, our research team is keen on retaining this analyzed section as part of our ongoing efforts to showcase the practical significance of this novel diagnostic agent in real-world medical scenarios.
- Your valuable input is much appreciated. We find the idea of conducting an ad-hoc analysis on the clinical application of FAPI PET/CT quite compelling. However, our research team is keen on retaining this analyzed section as part of our ongoing efforts to showcase the practical significance of this novel diagnostic agent in real-world medical scenarios.
- In conclusion, the work is adequately written but collects very inconsistent data. It is a collection of case studies that should be homogenized by talking about one topic at a time.
- In order to provide coherence, present, and discuss one topic at a time, we applied the following modifications:
- Most parts of the Abstract were re-written to introduce coherence with other parts (kindly track highlighted yellow text).
- Several missing findings in the Abstract was introduced to provide more comprehensive details matching those presented in other parts (gray text highlights).
- The Discussion section was redesigned utilizing subheadings (highlighted in yellow) to match and discuss all relevant findings presented in the results section and other sections. Kindly track the new text applied in the discussion and highlighted with gray color.
- The whole conclusion section was rewritten to introduce coherence with other parts (gray highlights).
- In order to provide coherence, present, and discuss one topic at a time, we applied the following modifications:

Reviewer 2 Report
Comments and Suggestions for Authors
In this manuscript, the authors compare various PET/CT methods that utilize different radiopharmaceuticals. While the paper presents some interesting findings, its structure could be improved. Specifically, there needs to be a more coherent connection among the abstract, results, discussion sections, and conclusions. The authors should enhance the presentation of their results, as some important findings are mentioned only briefly and lack detailed discussion. They raise several issues, but the main focus of the investigation remains unclear. In the discussion section, the authors note that their results are similar to those of other research groups. However, the manuscript should clarify its main contributions: what distinguishes this work from similar studies, and are there any notable differences from the findings of other scientific groups?
I appreciate the effort that has gone into this work, however, I believe the manuscript has significant potential for improvement. Additionally, some strong statements in the conclusions are not justified given the small sample size. As it stands, I am not convinced it warrants publication in its current form.
Author Response
Dear Reviewer 2
Thank you very much for this informative review shared along with respectful review points and vital questions that necessitate further improvement in many aspects of our article to cover each and every aspect of this vital subject.
Below are the answers to your respectful review points.
- In this manuscript, the authors compare various PET/CT methods that utilize different radiopharmaceuticals. While the paper presents some interesting findings, its structure could be improved. Specifically, there needs to be a more coherent connection among the abstract, results, discussion sections, and conclusions.
- In response to your respectful review point, we applied the following modifications:
- Most parts of the Abstract were re-written to introduce coherence with other parts (kindly track highlighted yellow text).
- Several missing findings in the Abstract was introduced to provide more comprehensive details matching those presented in other parts (gray text highlights).
- The Discussion section was redesigned utilizing subheadings (highlighted in yellow) to match and discuss all relevant findings presented in the results section and other sections. Kindly track the new text applied in the discussion and highlighted with gray color.
- The whole conclusion section was rewritten to introduce coherence with other parts (gray highlights).
- They raise several issues, but the main focus of the investigation remains unclear.
- This was further clarified in a supplementary table. Kindly track the applied changes highlighted in turquoise in lines 145-147 and review the supplemented table (Table S1).
- In the discussion section, the authors note that their results are similar to those of other research groups. However, the manuscript should clarify its main contributions: what distinguishes this work from similar studies, and are there any notable differences from the findings of other scientific groups?
- Thank you for pointing this out. Our study’s main contribution and key differences from previous ones were clarified in highlighted gray text in lines 370-374 and further detailed in the accompanying supplementary table (Table S3).
- I appreciate the effort that has gone into this work; however, I believe the manuscript has significant potential for improvement. Additionally, some strong statements in the conclusions are not justified given the small sample size.
- Conclusion was rewritten to lower the confidence tone and acknowledge the study limitations. Kindly track the highlighted gray text below conclusion subheading in lines 376-385.

Round 2
Reviewer 2 Report
Comments and Suggestions for Authors
This version of manuscript is improved and may be published in current form.